# Mepolizumab-Related Blood Eosinophil Decreases Are Associated with Clinical Remission in Severe Asthmatic Patients: A Real-World Study

**DOI:** 10.3390/antib14030061

**Published:** 2025-07-22

**Authors:** Matteo Bonato, Francesca Savoia, Enrico Orzes, Elisabetta Favero, Gianenrico Senna, Micaela Romagnoli

**Affiliations:** 1Pulmonology Unit, Ca’ Foncello Hospital, Azienda Unità Locale Socio-Sanitaria 2 Marca Trevigiana, 3100 Treviso, Italy; francesca.savoia@aulss2.veneto.it (F.S.); enrico.orzes@aulss2.veneto.it (E.O.); micaela.romagnoli@aulss2.veneto.it (M.R.); 2Allergological and Rare Disease Centre, Internal Medicine Department, Ca’ Foncello Hospital, Azienda Unità Locale Socio-Sanitaria 2 Marca Trevigiana, 31100 Treviso, Italy; elisabetta.favaro@aulss2.veneto.it; 3Allergy Unit and Asthma Center, Verona Integrated University Hospital, 37100 Verona, Italy; gianenrico.senna@univr.it

**Keywords:** asthma, eosinophils, blood, mepolizumab, remission

## Abstract

**Background**: Mepolizumab is an effective treatment for severe eosinophilic asthma, leading to a depletion of blood eosinophil levels, the clinical relevance of which remains unclear. **Objective**: The aim of this study was to assess the relationship between mepolizumab-induced blood eosinophil reduction and clinical outcome in patients with severe eosinophilic asthma, in particular, whether the magnitude of blood eosinophil reduction was associated with clinical remission. **Methods**: We conducted a real-world retrospective analysis of 58 adult patients with severe eosinophilic asthma treated with mepolizumab. Clinical and respiratory functional parameters were evaluated at the start of mepolizumab treatment (T0) and after two years of treatment (T2; mean follow-up: 22.8 ± 7.5 months). Blood eosinophil counts were recorded at T0 and during the first year of treatment (T1; mean follow-up: 7.7 ± 4.1 months). **Results**: After two years of mepolizumab treatment, 58 severe asthmatic patients showed significant improvements in ACT score, FVC, and FEV_1_ and a reduction in acute exacerbations and the use of maintenance therapies. Clinical remission was achieved in 55.1% of patients. Lower blood eosinophil counts during the first year (T1) were associated with greater improvements in lung function and fewer exacerbations. A greater relative decrease in eosinophils from baseline to T1 (ΔEOS%) was significantly associated with remission, reductions in exacerbations, and no maintenance OCS use. ΔEOS% was the only independent predictor of remission in the multivariate analysis. A ≥90% reduction predicted remission with 80% specificity (AUC = 0.726). **Conclusions**: Monitoring blood eosinophils after mepolizumab initiation could be a useful tool for predicting long-term response to treatment. In particular, a reduction by over 90% of peripheral blood eosinophils during the first year of mepolizumab treatment predicts clinical remission with a specificity of 80%. Considering the accessibility and the low cost of this biomarker, it may help to optimize long-term asthma management.

## 1. Introduction

Mepolizumab is a humanized monoclonal antibody targeting interleukin−5 (IL-5), a key cytokine involved in the growth, activation, and survival of eosinophils [1]. It is currently approved as an add-on maintenance therapy for patients with severe eosinophilic asthma [1,2], a subtype of asthma typically characterized by elevated levels of blood eosinophils and poor disease control despite high-dose inhaled corticosteroids.

Multiple randomized controlled trials (RCTs) and real-world evidence have consistently demonstrated the clinical efficacy of mepolizumab in this population. Mepolizumab has been shown to significantly reduce the rate of asthma exacerbations, decrease dependence on oral corticosteroids (OCSs), and improve both lung function and the control of asthma symptoms [1,2,3,4]. In addition, mepolizumab induces an immediate and sustained depletion of blood eosinophils: a significant reduction in circulating eosinophil counts can be observed as early as 24 h after the first dose [5,6] and is currently used as a marker of drug activity and patient adherence [1]. However, whether this depletion can be associated with long-term clinical outcomes—such as sustained improvement in symptom control, lung function, exacerbation rate, or clinical remission—remains unknown to this day.

In this study, we aimed to evaluate the relationship between mepolizumab-induced eosinophil reduction shortly after treatment initiation and long-term clinical outcome. Specifically, we sought to determine whether the magnitude of blood eosinophil depletion during the first year of treatment was associated with the achievement of clinical remission after two years of treatment.

## 2. Methods

### 2.1. Study Design

We performed a retrospective analysis of real-world clinical data of adults patients treated with mepolizumab for severe asthma in our outpatient clinic between 1 January 2018 and 1 March 2025 (Ospedale Ca’ Foncello, Treviso, Italy; project approved by local ethical committee ref.1307/CE Marca, RINOVA study).

### 2.2. Population

All patients had an asthma diagnosis according to GINA guidelines [2] and severe asthma according to the ERS/ATS consensus criteria [7]. All patients received mepolizumab for severe asthma as the first indication, and they were > 18 years old at treatment start. All patients who were treated complied with national indications for mepolizumab prescription: (a) blood eosinophils ≥150 cell/µL in the same year of treatment initiation and ≥300 cell/µL during the year before AND (b) at least 2 acute exacerbations treated for ≥ 3/days with systemic corticosteroids in the previous 12 months despite maximal inhalator therapy (GINA STEP 4–5) or (c) at least 6 continuous months of maintenance oral corticosteroid use reported in the previous 12 months [8]. All patients received 100 mg of mepolizumab every four weeks. An expert physician in severe asthma management made the decision to treat patients with mepolizumab rather than other biologic agents, based on clinical judgment. The following exclusion criteria were applied: (a) no blood eosinophil measurement within 90 days before starting mepolizumab; (b) no blood eosinophil measurement within 1 year after starting mepolizumab; (c) mepolizumab discontinued before the 2-year evaluation; and (d) prior treatment with other biologics for asthma.

### 2.3. Patient Evaluation and Variables Assessed

Due to the observational and real-life design of the study, patients were evaluated according to the internal protocol of our severe asthma center. No additional tests or evaluations were performed for research purposes. All severe asthma patients eligible for biologic therapy underwent an initial visit (T0) within 30 days prior to the first mepolizumab administration, often the same day as the first injection. The timing of subsequent visits was determined by the clinician on a case-by-case basis, according to each patient’s clinical characteristics. Clinical data and pulmonary function at treatment start (T0) and after two years of treatment (T2; mean follow-up: 22.8 ± 7.5 months) were extracted from patients’ electronic registry.

Blood eosinophil count was recorded within 90 days prior to treatment start and during the first year of treatment (T1; mean follow-up: 7.7 ± 4.1 months). Both absolute (EOS_T0_ and EOS_T1_) and relative (%EOS_T0_ and %EOS_T1_) eosinophil counts were considered. The relative change in eosinophil count from T0 to T1 (ΔEOS%) was calculated using the following formula:$$\Delta EOS\%=\frac{{EOS}_{T1}-{EOS}_{T0}}{{EOS}_{T0}}\times 100$$

Clinical data collected included smoking habits, atopy, asthma onset (>18 years), the number of acute exacerbations (AEs) in the previous year, asthma control test (ACT) scores, and maintenance anti-asthmatic treatment. Atopy was defined as elevated total serum IgE (>75 kU/L) combined with at least one instance of sensitization to a specific allergen. AEs were recorded based on patient self-report during follow-up visits and defined as episodes of acute worsening of asthma symptoms requiring a course of oral corticosteroids (OCSs) for ≥3 days. Maintenance anti-asthmatic prescriptions included the prescription of daily high-dose inhaled corticosteroids (ICSs) defined according to GINA guidelines, long-acting muscarinic antagonists (LAMAs), and oral corticosteroids (OCSs). Pulmonary function tests were conducted by expert pulmonary function technicians: pre-bronchodilator forced vital capacity (FVC) and pre-bronchodilator forced expiratory flow in the 1st second (FEV_1_) were considered to be variables in this study. Post-bronchodilator testing was not performed, as it was not part of the routine follow-up protocol.

Clinical remission was defined according to Menzies-Gow criteria [9] and operationalized as follows: (a) sustained absence of significant asthma symptoms based on the validated instrument, defined as an ACT score ≥ 20 at T2; (b) optimization and stabilization of lung function, defined as no decline in FEV_1_ between T0 and T2; (c) patient and healthcare provider agreement regarding remission; and (d) no use of systemic corticosteroid therapies for exacerbation treatments or long-term disease control.

### 2.4. Statistical Analysis

Patient characteristics were expressed as mean ± standard deviation (SD) for continuous variables, and as counts and percentages for categorical variables. Given the relatively small sample size, a non-normal distribution of continuous variables was assumed; therefore, non-parametric tests were used for statistical analyses.

Specifically, the Wilcoxon signed-rank test was used to assess differences in continuous variables over time, and the Mann–Whitney U test was used to compare independent groups. Spearman’s rank correlation coefficient was used to evaluate associations between continuous variables. Differences between categorical variables were assessed using the Chi-square test. A multivariate logistic regression analysis was performed to assess the association between eosinophil indices and clinical remission at T2. The *p*-values were derived from the Wald test. A receiver operating characteristic (ROC) curve analysis was also performed to evaluate the diagnostic accuracy of blood eosinophil reduction in predicting asthma remission at follow-up. The area under the curve (AUC) was used as a summary measure of test performance. All statistical analyses were conducted using IBM SPSS Statistics, version 23.0 (IBM Corp., Armonk, NY, USA). A *p*-value < 0.05 was considered statistically significant.

## 3. Results

Fifty-eight patients were included in the study (mean age: 66.2 ± 14.4 years, 55.1% female). The total population of patients with severe asthma who were followed up with in our outpatient clinic consisted of 140 subjects (mean age of 67.7 ± 15.9 years, 58.4% female), of whom 120 were receiving biologic therapy at the time. Demographic and clinical characteristics of patients at T0 and T2 are reported in Table 1.

A significant improvement was observed in all clinical and functional parameters that were evaluated after two years of treatment with mepolizumab. Specifically, mepolizumab proved to be effective in improving the asthma control test (ACT) score (a mean increase of 78.5 ± 57.2%; *p* < 0.0001), FVC (a mean increase of 11 ± 24.5%; *p* = 0.0134), and FEV_1_ (a mean increase of 17.7 ± 32.2%; *p* = 0.0002). Moreover, it significantly reduced the number of acute exacerbations (a mean reduction of 77.3 ± 53.5%; *p* < 0.0001) as well as the prevalence of patients on maintenance OCS treatment (a reduction of 66%; *p* = 0.0008), high-dose ICS (a reduction of 29.6%; *p* = 0.0341), and LAMA (a reduction of 65.6%; *p* = 0.0011). Notably, 32 patients (55.1%) achieved clinical remission according to the Menzies-Gow criteria.

As shown in Figure 1a, a significant reduction in both absolute and relative blood eosinophil count was observed from T0 (mean 1011.6 ± 1446.1 cell/µL, 9.5 ± 6.6%) to T1 (mean 97.5 ± 87.9 cell/µL, 1.5 ± 1.1%), with a mean relative reduction (ΔEOS%) of −83.9 ± 12.3% (*p* < 0.0001).

When exploring the correlations between blood eosinophil counts and changes in clinical–functional outcomes, no associations were observed between either absolute or relative eosinophil counts at baseline (EOS_T0_ and %EOS_T0_) and any clinical–functional improvement or achievement in clinical remission at two years (T2).

Conversely, lower absolute blood eosinophil counts at T1 (EOS_T1_) were weakly but significantly correlated with better clinical and functional improvements from T0 to T2. In particular, a lower EOS_T1_ was associated with greater improvements in pulmonary function (*p* = 0.009, r = −0.423 for FVC, Figure 1b; *p* = 0.003, r = −0.282 for FEV_1_, Figure 1c) and with greater reductions in acute exacerbations (*p* = 0.034, r = 0.291, Figure 1d) from T0 to T2. Furthermore, both EOS_T1_ and %EOS_T1_ counts were significantly lower in patients who achieved clinical remission at T2 (76.5 ± 56.4 cell/µL vs. 123.4 ± 110.8 cell/µL, *p* = 0.042, and 1.0 ± 0.95% vs. 1.8 ± 1.2%, *p* = 0.0013; Figure 1e,f). Notably, no significant associations were observed between blood eosinophil counts at T1 and maintenance therapy at T2.

When extending our analysis to include the relative change in eosinophil count from T0 to T1 (ΔEOS%), a greater decrease in blood eosinophils from T0 to T1 was moderately associated with a greater reduction in the number of acute exacerbations observed from T0 to T2 (*p* = 0.0018; r = 0.428; Figure 2a). Moreover, ΔEOS% was significantly associated both with no OCS chronic treatment (–86 ± 11.3% vs. −75 ± 13%, *p* = 0.009; Figure 2b) and with clinical remission at T2 (–88.2 ± 101% vs. –78.8 ± 71%, *p* = 0.004; Figure 2c).

A multivariate logistic regression model was performed to assess the relationship between eosinophil indices predictive of clinical remission at T2 in univariate models. Variables included in the model were ΔEOS%, EOS_T1_, and %EOS_T1_. The model was statistically significant (*p* = 0.007; R^2^ = 0.164), and among the predictors tested, only ΔEOS% was independently associated with remission at T2 (OR = 0.912; 95% CI: 0.844–0.986; *p* = 0.02), while EOS_T1_ and %EOS_T1_ were not significant (Table 2).

The association between ΔEOS% and clinical remission at T2 remained statistically significant even after stratifying the sample by the time interval between T0 and T1: <6 months (−76.3 ± 16.6% vs. −88.3 ± 10.8%, *p* = 0.031) and >6 months (−80.2 ± 11% vs. −88 ± 9.5%, *p* = 0.049). Finally, the receiver operating characteristic (ROC) analysis (Figure 2d) showed a moderate predictive accuracy of ΔEOS% for clinical remission at T2 (AUC = 0.726), with a sensitivity of 52% and a specificity of 80% at a threshold of −90%.

## 4. Discussion

In our study, for the first time, we demonstrated the potential utility of monitoring blood eosinophils in mepolizumab treatment during the follow-up assessments of severe asthmatics. Indeed, in our cohort of severe asthmatic patients who were biologically naïve and treated with mepolizumab, an early blood eosinophil reduction predicts long-term clinical outcome with moderate accuracy. In particular, we demonstrated that lower levels of blood eosinophils after treatment initiation were associated with a 2-year pulmonary function improvement and reduction in acute exacerbations. Furthermore, a greater reduction from baseline after treatment initiation (ΔEOS%) was significantly associated with positive outcomes such as reductions in acute exacerbations, weaning from OCSs, and clinical remission after two years of treatment. Conversely, similar correlations have not been observed for baseline blood eosinophil count.

Due to the registration of randomized controlled trials, it is now well established that high peripheral blood eosinophil counts at baseline are associated with a better response to mepolizumab [1,3]. However, there is more limited evidence regarding the linearity of this association for values above 300 cells/cell/µL [3,10,11]. More specifically, a post hoc meta-analysis by Albers et al. based on MENSA and MUSCA trials shows that the linear relationship between clinical response and baseline eosinophil levels tends to be lost at very high eosinophil counts (above 500 cells/µL), particularly with regard to the reduction in exacerbations and improvement in FEV_1_ [12]. Therefore, within our selected population of eosinophilic asthmatics—all of whom, due to mepolizumab prescription restrictions, have baseline eosinophil counts above 300 cells/µL—the eosinophil levels and the magnitude of eosinophil depletion following treatment initiation (ΔEOS%), rather than baseline eosinophil counts alone, represent predictors of long-term clinical response. More precisely, we proved that a reduction of more than 90% in blood eosinophil count from baseline during the first year of treatment (ΔEOS%) was associated with clinical remission during the second year of treatment with a specificity of 80%, despite low sensitivity. Translated into practical terms, when an over 90% decrease in eosinophils is observed after mepolizumab administration, it is possible to speculate on future clinical remission; conversely, no such inference can be made when the reduction is below 90%. Moreover, we specify that this potential clinical usefulness of blood eosinophils is limited to anti-IL-5 antagonists, as targeting IL-5R generally zeroes out peripheral eosinophils [2].

A previous study by Mukherjee et al. [13] in a larger population concluded that blood eosinophil monitoring after initiating anti-IL5 therapy was not useful to identify suboptimal responders, although their criteria for defining non-response were different from ours, as our follow-up timing was not the same as theirs.

We are aware that our results have several limitations, primarily related to the non-standardized timing of eosinophil monitoring, which varies from patient to patient. We fully acknowledge that this represents a potential source of bias, as it is well known that peripheral blood eosinophils can show substantial intra-individual variability over time [14]. Remarkably, this aspect mirrors the real-life clinical practice, where the timing of the blood sample is largely influenced by patient availability. However, this temporal variability did not appear to significantly affect the validity of ΔEOS% as a predictive biomarker of clinical remission in our cohort.

## 5. Conclusions

In conclusion, our findings suggest that monitoring blood eosinophils after the first administration of mepolizumab could be a useful tool for predicting long-term responses to treatment. In particular, an early reduction by over 90% of peripheral blood eosinophil after mepolizumab treatment initiation (ΔEOS%) predicts long-term clinical remission with a low sensitivity but high specificity. We are aware that the association we found is modest, but considering the low cost of the test and its accessibility, we consider it potentially useful in routine clinical practice and deserving of being validated by future prospective studies in larger cohorts by standardizing the timing of blood samples.

## Figures and Tables

**Figure 1 antibodies-14-00061-f001:**
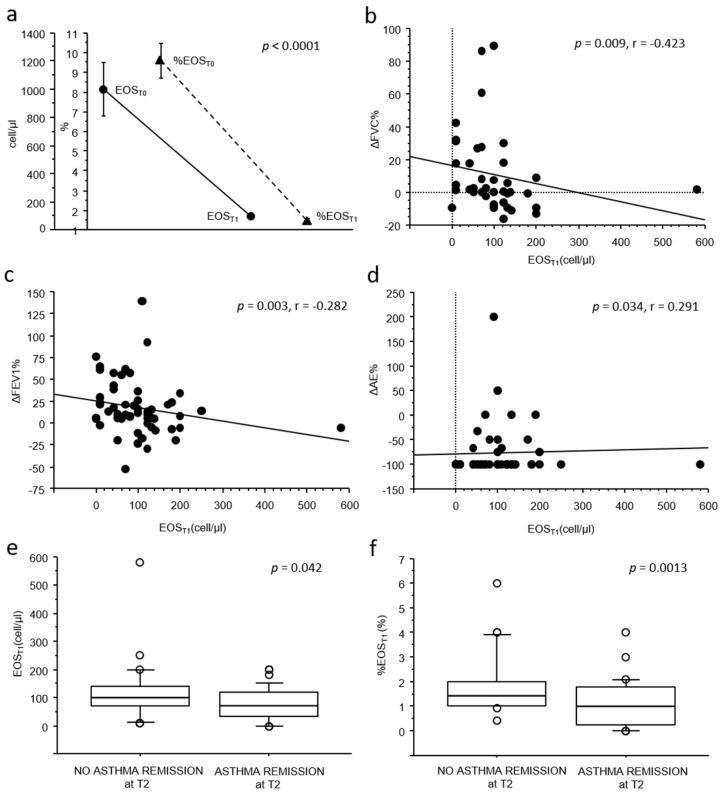
(**a**) Line chart showing means of EOS_T0_ and EOS_T1_ (dots), as well as %EOS_T0_ and %EOS_T1_ (triangles); error bars are represented as vertical bars. Differences between time points were evaluated using the Wilcoxon signed-rank test; (**b**–**d**) bivariate scatterplots showing the correlation (Spearman’s rank test) between blood eosinophil count at T1 (EOS_T1_) and the relative variation from T0 to T2 in FVC (∆FVC%), FEV_1_ (∆FEV_1_%), and the number of acute exacerbations in the previous year (∆AE%); (**e**,**f**) box plots showing the difference in EOST1 and %EOST1 according to clinical remission at T2 (Mann–Whitney U-test). The solid line represents the median; the bottom and top of the boxes are the 25th and 75th percentiles; and the brackets correspond to the 10th and the 90th percentiles.

**Figure 2 antibodies-14-00061-f002:**
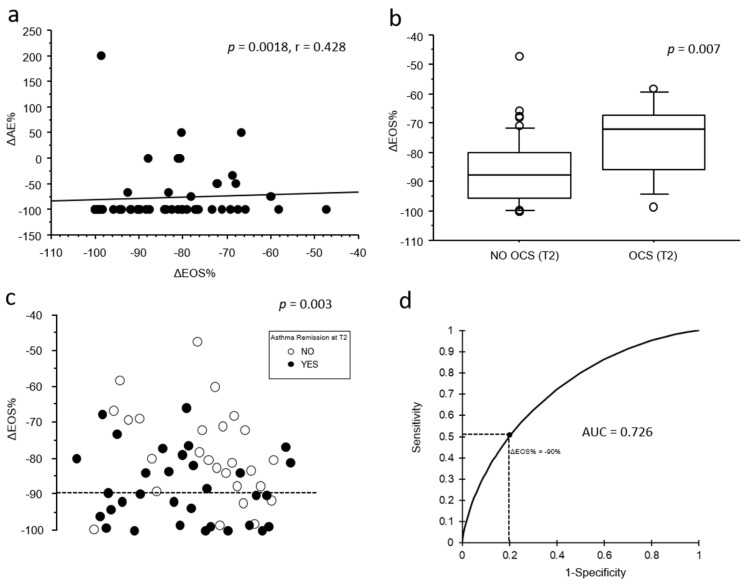
(**a**) Bivariate scatterplots showing the correlation (Spearman’s rank test) between the relative variation in blood eosinophil from T0 to T1 (ΔEOS%) and the relative variation in the number of acute exacerbations in the previous year (∆AE%); (**b**) box plot showing the difference in ΔEOS% according to the persistence of OCS maintenance therapy at T2 (Mann–Whitney U-test); (**c**) univariate scatterplot reporting the distribution of the relative blood eosinophil variation from T0 to T1 (ΔEOS%) according to asthma clinical remission at T2 (Mann–Whitney U-test). White dots represent patients who achieved clinical remission at T2, while black dots represent those who did not. The dashed bar represents the ROC extrapolated threshold of −90%; (**d**) receiver operating characteristic (ROC): bivariate line chart representing the variation in 1-specificity (X axis) and sensitivity (Y axis) accordingly to different values of (ΔEOS%) in predicting clinical remission at T2. The best cut-off of ΔEOS% = −90% is represented as dashed lines.

**Table 1 antibodies-14-00061-t001:** Clinical and demographic characteristics of patients at baseline (T0) and follow-up (T2).

	T0	T2	*p*-Value
Number of patients	58	58	-
Age (y)	66.2 ± 14.4	68.1 ± 14.2	-
Never smokers, *n* (%)	13 (22.5)	-	-
Former smokers, *n* (%)	45 (77.5)	-	-
Sex female, *n* (%)	32 (55.1)	-	-
Atopics, *n* (%)	16 (27.5)	-	-
Early-onset asthmatics, *n* (%)	5 (8.8)	-	-
ACT score, pts	12.9 ± 3.9	21.3 ± 3.8	<0.0001
FVC, L	3.1 ± 1.2	3.4 ± 1.2	0.0134
FEV_1_, L	1.9 ± 0.8	2.2 ± 0.9	0.0002
AEs in previous year, *n*	2.3 ± 1.5	0.4 ± 0.8	<0.0001
Maintenance therapy:			
OCSs, *n* (%)	32 (55.1)	11 (18.9)	0.0008
High-dose ICSs, *n* (%)	54 (93.1)	38 (65.5)	0.0341
LAMAs, *n* (%)	19 (32.7)	10 (17.2)	0.0011
Remission achieved, *n* (%)	n.a.	32 (55.1)	-

Data are expressed using the mean ± SD for continuous variables, and counts and percentages for nominal variables. ACT = asthma control test; FVC = forced vital capacity; FEV_1_ = forced expiratory volume in 1 s; AEs = acute exacerbations; OCSs = oral corticosteroids; ICSs = inhaled corticosteroids; LAMAs = long-acting anti-muscarinic antagonist; n.a. = not applicable. *p*-values were obtained from the Wilcoxon signed-rank and Chi-square tests for continuous and categorical variables, respectively.

**Table 2 antibodies-14-00061-t002:** Multivariate logistic regression model predicting clinical remission at T2.

	OR	95%CI	*p*-Values
**Dependent variable:**			
Clinical remission at T2: yes			
**Independent variables:**			
ΔEOS% (%)	0.912	0.844–0.986	0.02
EOS_T1_ (cell/µL)	1.013	0.996–1.03	0.1443
%EOS_T1_ (%)	0.375	0.135–1.043	0.0603

Multivariate logistic regression R^2^ = 0.164.

## Data Availability

The data that support the findings of this study are available from the corresponding author, M.B., upon reasonable request.

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
