# Peer review of "Mepolizumab-Related Blood Eosinophil Decreases Are Associated with Clinical Remission in Severe Asthmatic Patients: A Real-World Study"

_2073-4468, 2025, doi:10.3390/antib14030061_

Round 1
Reviewer 1 Report
Comments and Suggestions for Authors
Mepolizumab-related blood eosinophil decrease is associated with clinical remission in severe asthmatic patients
First of all, the study is very interesting and relevant. However, I would like to recommend a few changes before it proceeds to the next stage:
- Title: It would be better to indicate in the title that this is a real-world observation or real-world data analysis.
- Abstract: The r-correlation is not the appropriate tool for assessing a relationship; please clarify this.
- Methods section: Was this a sample or the entire population? Please specify.
- Statistical analysis section: Use “differences” instead of “distributions” where appropriate. Also, please describe how the distribution of quantitative variables was tested, since the sample size is small and it is likely that the distribution deviates from normality.
- Figure 1: The authors present a correlation analysis, but no correlation analysis is described in the Methods section. Please clarify which correlation test was used. For testing differences in repeated measures of asymmetrically distributed variables, the Wilcoxon test should be used.
- Table 1: This table should be recalculated. In the Results section, please add a note in the table footnotes or in an additional column indicating which statistical test was used for each comparison.
- Figure 1a: Please specify which statistical test was used.
- Logistic regression: Logistic regression should be used to calculate odds ratios (OR), not risk ratios (RR). Please recalculate this analysis and clearly specify the outcome variable and the independent variables included in the model in the methods section, as well as the rationale for their inclusion. Also, please add a separate table presenting the logistic regression results, including OR, 95% CI, and p-values.
Author Response
First of all, the study is very interesting and relevant. However, I would like to recommend a few changes before it proceeds to the next stage:
We are grateful to the reviewer for this appreciation. We provided to answer to his/her observations:
R1.1: It would be better to indicate in the title that this is a real-world observation or real-world data analysis.
A1.1: we appreciate this suggestion, we specified it in the title.
R1.2: The r-correlation is not the appropriate tool for assessing a relationship; please clarify this.
A1.2: we thank the reviewer for this observation, we clarified that it was not an r-correlation but a Spearman’s rank correlation in Methods section (line 136).
R1.3: Was this a sample or the entire population? Please specify.
A1.3: yes, it was a sample of a population of severe asthmatic, we clarified this point in the manuscript (line 149-151)
R1.4-7: Statistical analysis section: Use “differences” instead of “distributions” where appropriate. Also, please describe how the distribution of quantitative variables was tested, since the sample size is small and it is likely that the distribution deviates from normality.
Figure 1: The authors present a correlation analysis, but no correlation analysis is described in the Methods section. Please clarify which correlation test was used. For testing differences in repeated measures of asymmetrically distributed variables, the Wilcoxon test should be used.
Table 1: This table should be recalculated. In the Results section, please add a note in the table footnotes or in an additional column indicating which statistical test was used for each comparison.
Figure 1a: Please specify which statistical test was used.
A1.4-7: The reviewer is absolutely right: given the small sample size, it is appropriate to assume non-normality of continuous variables. We have therefore clarified in more detail the statistical tests used in the 'Statistical Analyses' section and used the term differences instead of distribution (line 130-145). We recalculated the p-values using non-parametric tests: the Wilcoxon rank-sum test for repeated measures and the Mann–Whitney U test for comparisons between independent groups and we did not observe any loss of statistical significance. The updated p-values and the statistical tests applied are now clearly specified in Table 1 and in figures. We also confirm that correlations between continuous variables had already been calculated using non-parametric test (Spearman’s rank).
R1.8: Logistic regression: Logistic regression should be used to calculate odds ratios (OR), not risk ratios (RR). Please recalculate this analysis and clearly specify the outcome variable and the independent variables included in the model in the methods section, as well as the rationale for their inclusion. Also, please add a separate table presenting the logistic regression results, including OR, 95% CI, and p-values.
A1.8: We thank the reviewer for this observation. We provided to clarify this point and we added a new table with multivariate results (line 200-206, table 2).
Reviewer 2 Report
Comments and Suggestions for Authors
Thank you for your manuscript submission. It is a manuscript with insights about the treatment of asthma patients with mepolizumab.
However, I have some questions:
- Maybe creating or suggesting a possible standard procedure of evaluating this kind of patient should be proposed in the discussion, meaning that it is hard to evaluate the interim assessment of blood eosinophils and lung function when the T0 and T1 time varies so much. Maybe it would be interesting to include patients within 1 month, 3 months, 6 months, and 1 year visits after the first dose of mepolizumab?
- Have you considered taking an allergic asthma group separately and evaluating them?
Author Response
Thank you for your manuscript submission. It is a manuscript with insights about the treatment of asthma patients with mepolizumab.
We thanks the reviewer for the review of our manuscript. We provided to answer to his/her observations:
R2.1: Maybe creating or suggesting a possible standard procedure of evaluating this kind of patient should be proposed in the discussion, meaning that it is hard to evaluate the interim assessment of blood eosinophils and lung function when the T0 and T1 time varies so much. Maybe it would be interesting to include patients within 1 month, 3 months, 6 months, and 1-year visits after the first dose of mepolizumab?
A2.1: We thank the reviewer for his/her valuable suggestion. We agree that having a standardized approach for timing the interim assessments of blood eosinophils and lung function would improve the evaluation of patients, especially considering the variability in timing between T0 and T1 also. However, this variability reflects the real-world clinical practice where sample timing depends from patients’ availability. Unfortunately, in our study it was not possible to perform stratified analyses based on the timing of sample collection suggested by the reviewer due to the limited sample size. However, we provided a stratified analysis based on the time interval between T0 and T1 (>6 months vs <6 months), and found that ΔEOS% remained significantly associated with clinical remission at 2 years (line 209-212) in both groups. We have addressed this point in the Discussion section both as a possible source of bias (line 264-267) as a suggestion for further studies (line 279)
R2.2: Have you considered taking an allergic asthma group separately and evaluating them?
A2.2: We thank the reviewer for his/her insightful question. Sixteen patients in our cohort were defined as atopic (Table1). However, when analyzing this subgroup separately, the primary outcome of the study was not confirmed among atopic patients alone. We believe this result is largely influenced by the small sample size in this small subgroup (9 patients in remission vs. 7 not in remission).
Reviewer 3 Report
Comments and Suggestions for Authors
Dear Authors,
I have read your manuscript regarding a retrospective observational study on the relationship between the reduction in peripheral blood eosinophils after initiating mepolizumab therapy and long term clinical outcomes in patients with severe eosinophilic asthma. Your study is well structured, statistically sustained and seems relevant for clinical practice.
The use of delta-EOS% as a biomarker for predicting long-term remission increases validity and sustains clinical relevance, but although the biomarker use is clearly explained, the handling of missing eosinophil data is not clearly discussed and addressed. For example, the timing of T1 blood tests is presented as highly variable (with a mean 7.7 ± 4.1 months). This is not clarified in terms of bias, so please clarify this aspect.
Also, as you discuss "clinical remission", please provide the full criteria with precise attribution to the original Menzies-Gow definition.
I have understood that the delta-EOS% cut-off for predicting remission is at 52% (low sensitivity). I encourage you to highlight and discuss this, despite statistical significance.
Also, the argument that monitoring eosinophils could guide treatment switch is interesting, but speculative. I recommend tempering this claim unless you provide literature data that can support it.
Please check and correct the English language for clarity, avoid confusing phrasing, and the use of incorrect verb tenses.
I am willing to review again after the authors make proper modifications and clarifications.
Comments on the Quality of English LanguagePlease check English language. I am not an Engllish native speaker but I detected some awkward phrasing and use of tenses:
for example, in the abstract you write: " Clinical relevance of this eosinophil reduction is remains unclear. " (row 15) or "Post-bronchodilator tests was not been performed because not included in the routine clinical assessment of patients in follow-up." (row 115). Please also check spelling and verb-subject usage.
Also, I did not understand what you meant by: "Moreover, a significant more marked decrease of blood eosinophils from T0 to T1" (row 180) and "weaning from OCS and achieving of clinical remission at T2" (row 182). Please rephrase for clarity, use the appropriate grammar and avoid confusional phrasing such as the one described above.
Author Response
I have read your manuscript regarding a retrospective observational study on the relationship between the reduction in peripheral blood eosinophils after initiating mepolizumab therapy and long term clinical outcomes in patients with severe eosinophilic asthma. Your study is well structured, statistically sustained and seems relevant for clinical practice.
We thank the reviewer for his/her appreciations. We provided a full response to his/her comments.
R3.1: The use of delta-EOS% as a biomarker for predicting long-term remission increases validity and sustains clinical relevance, but although the biomarker use is clearly explained, the handling of missing eosinophil data is not clearly discussed and addressed. For example, the timing of T1 blood tests is presented as highly variable (with a mean 7.7 ± 4.1 months). This is not clarified in terms of bias, so please clarify this aspect.
R3.1: We thank the reviewer for this insightful comment, which allowed us to clarify and strengthen the methodological rigor of our analysis. Regarding missing eosinophil data, we would like to clarify that no data were missing for this variable, as the availability of both T0 and T1 eosinophil counts was an explicit inclusion criterion for the study. Regarding the time variability of the T1 blood sample, we fully acknowledge that this represents a potential source of bias, as it is well known that peripheral blood eosinophils can show substantial intra-individual variability over time. However, in routine clinical practice, standardizing blood sample timing is often not feasible. This aspect has been discussed in the manuscript (line 264-271). To address the Reviewer’s concern more directly, we stratified the cohort based on whether the T1 sample was obtained before or after 6 months from T0. The association between delta-EOS% and clinical remission remained statistically significant in both subgroups (line 209-211). Once again, we are grateful to the Reviewer for raising this important point, which helped us improve the clarity and strength of our results.
R3.2: Also, as you discuss "clinical remission", please provide the full criteria with precise attribution to the original Menzies-Gow definition.
A3.3: We thank the reviewer for the opportunity to clarify this important point. We have now reported the original criteria for clinical remission exactly as stated in reference 9 (line 122-127). Criteria (c) and (d) are unambiguous. Criterion (a) was interpreted as an ACT score ≥ 20, which is also acknowledged as acceptable in reference 9. Criterion (b) is not defined operationally in the original Menzies-Gow criteria; therefore, we interpreted it as the absence of FEV₁ decline over a 2-year period.
R3.3: I have understood that the delta-EOS% cut-off for predicting remission is at 52% (low sensitivity). I encourage you to highlight and discuss this, despite statistical significance. Also, the argument that monitoring eosinophils could guide treatment switch is interesting, but speculative. I recommend tempering this claim unless you provide literature data that can support it.
A3.3: We thank the reviewer for this valuable comment. We have revised the discussion to highlight the low sensitivity of the delta-EOS% cut-off (line 255-257). We fully agree that the argument regarding eosinophil monitoring as a tool to guide treatment switching is highly speculative and not well supported by the current literature. Therefore, we have chosen to remove this statement and instead provide a more balanced practical implication.
R3.4: Please check and correct the English language for clarity, avoid confusing phrasing, and the use of incorrect verb tenses.
R3.4: We thanks the reviewer for this observation. The manuscript has been reviewed by English language experts among the co-authors. We thank the reviewer for pointing this out.
Round 2
Reviewer 3 Report
Comments and Suggestions for Authors
The authors made the required modifications and the article can be accepted as is